# Feasibility of Using Laser Imaging Detection and Ranging Technology for Contactless 3D Body Scanning and Anthropometric Assessment of Athletes

**DOI:** 10.3390/sports12040092

**Published:** 2024-03-26

**Authors:** Katja Oberhofer, Céline Knopfli, Basil Achermann, Silvio R. Lorenzetti

**Affiliations:** 1Section Performance Sport, Swiss Federal Institute of Sport Magglingen (SFISM), Hauptstrasse 247, 2532 Magglingen, Switzerland; katjaoberhofer@outlook.com (K.O.); celine.knopfli@hest.ethz.ch (C.K.); basil.achermann@baspo.admin.ch (B.A.); 2Department of Health Sciences and Technology (HEST), ETH Zurich, LFW C13.2, Universitätsstrasse 2, 8092 Zurich, Switzerland

**Keywords:** anthropometry, LiDAR, 3D shape analysis, body sizing, strength training, sports science

## Abstract

The scope of this pilot study was to assess the feasibility of using the laser imaging detection and ranging (LiDAR) technology for contactless 3D body scanning of sports athletes and deriving anthropometric measurements of the lower limbs using available software. An Apple iPad Pro 3rd Generation with embedded LiDAR technology in combination with the iOS application Polycam were used. The effects of stance width, clothing, background, lighting, scan distance and measurement speed were initially assessed by scanning the lower limbs of one test person multiple times. Following these tests, the lower limbs of 12 male and 10 female participants were scanned. The resulting scans of the lower limbs were complete for half of the participants and categorized as good in quality, while the other scans were either distorted or presented missing data around the shank and/or the thigh. Bland–Altman plots between the LiDAR-based and manual anthropometric measures showed good agreement, with the coefficient of determination from correlation analysis being R^2^ = 0.901 for thigh length and R^2^ = 0.830 for shank length, respectively. The outcome of this pilot study is considered promising, and a further refinement of the proposed scanning protocol and advancement of available software for 3D reconstruction are recommended to exploit the full potential of the LiDAR technology for the contactless anthropometric assessment of athletes.

## 1. Introduction

Subject-specific anthropometric measurements are needed in a broad context across ergonomics, engineering, design research, health and sports sciences. Therefore, subject-specific body measurements are traditionally obtained by hand according to the standards of the International Society for the Advancement of Kinanthropometry (ISAK) and used for, e.g., designing new customer goods, assessing patient characteristics or monitoring training progress [1].

In recent years, automatic 3D scanners have provided new means to capture body surface data of individual subjects, and these are contactless, with high repeatability and speed [2]. Particularly, a new light detection and ranging (LiDAR) sensor for depth sensing was introduced in 2020 by Apple (Apple Inc., Cupertino, CA, USA) into their high-end mobile devices, which has opened the way for convenient 3D scanning outside the laboratory. The LiDAR technology works by emitting arrays of infrared light pulses from a series of transmitters into the environment, which are reflected from the surface of the target object and re-captured by integrated photodetector sensors. The sensors detect the frequency of the reflected light, which is then used to calculate travel time and distance to the target surface [2]. 

The Apple LiDAR technology has been adopted for, e.g., forensic 3D documentation [3], large animal assessments in agriculture [4] and to estimate tree diameter in the context of forest management [5]. Furthermore, 3D scanning using commercial mobile devices has proven useful for preoperative and postoperative analysis of facial structures by plastic surgeons [6], as well as the estimation of body segment parameters for biomechanical analysis [7]. Yet, the potential of the Apple LiDAR technology for 3D body scanning and anthropometric assessment of sports athletes has not yet been demonstrated. 

The goal of this pilot study was to assess the feasibility of using the Apple LiDAR technology for contactless anthropometric measurements of strength-training athletes outside the dedicated laboratory and to assess the feasibility of extracting anthropometric measures from the 3D data using available iOS software (Version 15.5). It was hypothesized that the LiDAR technology allows for the contactless measurement of shank and thigh length based on 3D body surface scanning in a training-specific setting with manual measurements, according to ISAK standards as reference values. 

## 2. Materials and Methods

Ethical approval for this study was given by the regional ethics committee (Kantonale Ethikkommission Bern, Nr: 2021-00403). A total of 22 healthy, recreationally active subjects (*n* = 12 M/10 F, age = 29 ± 4.7, height = 1.64 ± 0.38 m; body mass = 76 ± 12 kg) gave written informed consent to participate in this study. An Apple iPad Pro 3rd Generation with embedded LiDAR technology was used in combination with the iOS application Polycam for 3D scanning, visualization and analysis (https://poly.cam, accessed on 1 September 2022). 

The effect of leg distance, clothing, background, lighting, scan distance and measurement speed were initially assessed by scanning the lower limbs of one test person multiple times. Based on this initial testing, the measurement protocol for scanning the lower limbs was defined as follows: (1) uniform background and lighting, (2) participant wearing tight, single-coloured clothing or only presenting naked skin, (3) participant standing in a T-position with standardized leg distance of 25–30% body height, (4) examiner moving at constant, moderate speed on a circular path around the target, (5) keeping the iPad as stable as possible, perpendicular to the plane of motion and (6) keeping a constant distance of 50–100 cm from the target.

Each subject was scanned according to the above protocol, whereby all scans were performed by one examiner. The same examiner also performed the test scans in order to familiarize herself with the technology. Additionally, anthropometric data of all participants, including size, weight, thigh and shank lengths and circumferences, were manually measured by a trained practitioner according to ISAK standards. Each measure was taken twice to be averaged. Data acquisition was not randomised. For all participants, the LiDAR scan was firstly obtained, followed by manual anthropometric measurements. 

The 3D models from LiDAR scanning were visually assessed and analysed using the iOS application Polycam. Measurements of thigh and shank length were extracted as study outcome parameters from the 3D models using the integrated linear measuring tool. Thus, 3D models were categorized into ‘poor’, ‘moderate’ and ‘good’ depending on the completeness of body surface data. Particularly, 3D models were categorized as ‘poor’ if the extraction of thigh and shank lengths was not possible and as ‘moderate’ if only thigh or shank length could be individually extracted.

The length measures from the left and right leg of all participants were combined and statistically compared between the manual and the LiDAR-based data. The comparison was limited to thigh and shank length measurements due to the constraints of the Polycam software (Version 3.2.7), which only allowed for linear measurements to be extracted. Prior to statistical analysis, data were checked for normal distribution using the Shapiro–Wilk test. Student’s paired *t*-tests were then used to determine whether the differences in lengths measures between the manual and the LiDAR-based measures were statistically significant, with the level of significance set at *p* < 0.05. Furthermore, the correlation and agreement between the length measures from manual versus LiDAR-based assessment were analysed by calculating the coefficient of determination (R^2^) and visualizing the data using Bland–Altman plots, with the confidence interval set at 95% limits of agreement [8].

## 3. Results

The data of all the 22 participants were included in the evaluation. The average thigh and shank lengths of both legs from the manual and the LiDAR-based assessments including statistical results are given in Table 1. Thus, the scans of six participants were either too distorted or contained missing data, thereby unable to extract shank and thigh lengths (i.e., poor scans), and the scans of another five participants only allowed thigh length measures to be derived (i.e., moderate scans). The scans of eleven participants were categorized as good, allowing for the extraction of shank and thigh lengths of both legs from the 3D point clouds. Consequently, this categorization yielded 16 thigh length measurements (from the ‘moderate’ and ‘good’ groups, n = 5 + 11) and 11 shank length measurements (from the ‘good’ group, n = 11). A representative sample of LiDAR scans is given in Figure 1, and the results from the correlation analysis and Bland–Altman plots are shown in Figure 2. 

## 4. Discussion

Given the growing popularity of LiDAR technology as a consumer electronic device, the goal of this pilot study was to provide guidelines and praxis-oriented insights into the potential of the Apple LiDAR technology for convenient anthropometric assessment in a sport-specific setting. Based on initial testing, the measurement protocol was defined to ensure uniform background and lighting, with the examiner moving at a constant moderate speed around the subject and keeping the iPad stable and perpendicular to the plane of motion at a constant distance to the target. Nevertheless, half of the resulting scans were only moderate or poor in quality, with distortions or missing data especially between the legs and closer to the floor (Figure 1). 

Inconsistent lighting between the legs, as well as around the shank and ankle close to the floor, may have contributed to the poor 3D reconstruction in these areas. Unfortunately, no decisive conclusion could be drawn regarding the best choice of garment, colour and/or bare skin to improve scan quality. In similar work of facial scanning, it was also found that areas with inconsistent lighting and increased specular reflectivity (e.g., nose and chin) led to higher inaccuracies [6]. Thereby, the influence of skin type on scan outcome was also inconsistent in previous work [6]. Further experiments with additional adjustments to the present scan protocol are thus highly recommended, including other software packages for reconstruction. 

The quality of scans using LiDAR technology is largely dependent on the 3D reconstruction capability of the chosen software. In the present work, the iOS application Polycam was used for 3D scanning and data visualization. Unfortunately, the integrated software tool only allowed for the extraction of linear measurements from the 3D point cloud, i.e., shank and thigh lengths but not circumferences. There are a few fitness-specific stationary 3D scanners available, as well as mobile applications to estimate body dimensions based on RGB images from different views [2]. Yet, to the authors knowledge, there is no software available for anthropometric assessment based on LiDAR data. Further software development is highly encouraged and may likely help to improve analysis results.

This pilot study was part of a larger study to improve the safety and efficiency of strength training by means of mobile technology [9,10]. These results indicate a degree of reliability (Table 1), but further validation studies with a repeated measurement design and larger sample sizes are needed to draw decisive conclusions regarding the validity and reliability of LiDAR technology for contactless anthropometric assessment. The potential advantage of using 3D scanning technology compared to manual anthropometric assessment is the measurement speed, with the duration of a scan taking less than 1 min, as well as the possibility for the layperson to obtain an accurate measurement of body dimensions without prior training. Additionally, larger and more diverse body scanning datasets may become publicly available, with advances in deep learning algorithms and optimization techniques as well for the improved monitoring of physical training and rehabilitation progress. 

## 5. Conclusions

Advancements in LiDAR technology, as embedded in mobile devices, are opening the doors for convenient 3D body surface scanning and anthropometric assessment in a sport-specific setting. Despite challenges with inconsistent lighting across body parts and remaining software limitations, the outcomes of this pilot study are considered promising. Further advancements of the proposed scanning protocol and available software for 3D reconstruction are highly recommended to exploit the full potential of the LiDAR technology. For validation purposes, future studies should consider a repeated measurement design with larger sample sizes to substantiate the present preliminary results with scientific rigor. The ability to conveniently assess subject-specific body dimensions using mobile devices outside the dedicated laboratory is expected to help in the monitoring of training and rehabilitation outcomes to the benefit of athletes and patients alike. 

## Figures and Tables

**Figure 1 sports-12-00092-f001:**
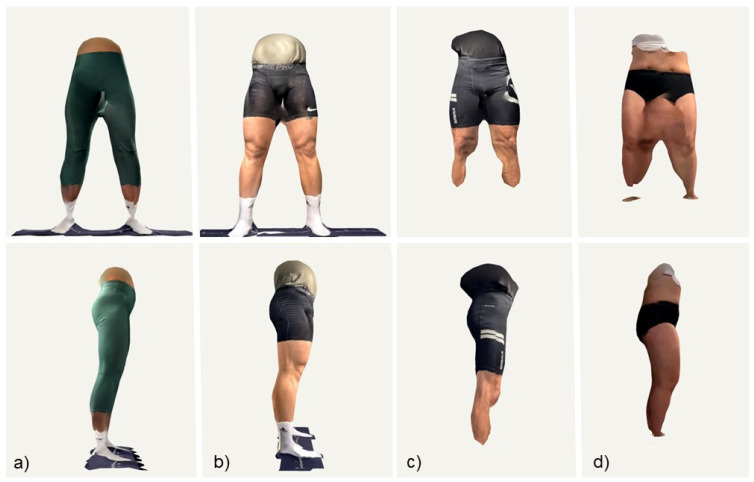
Representative sample of LiDAR scans of the lower limbs from frontal (**top row**) and side view (**bottom row**). (**a**) Good scan of female participants, (**b**) good scan of male participant, (**c**) incomplete scan of male participant, (**d**) incomplete and distorted scan of female participant. Visualisation and stillshots of the 3D point clouds were done using the iOS application Polycam.

**Figure 2 sports-12-00092-f002:**
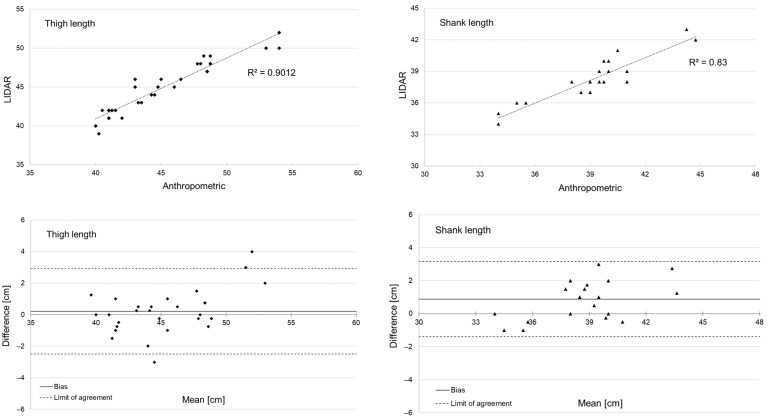
Correlation (**top**) with coefficient of determination (R^2^) and Bland–Altman plots (**bottom**) with the confidence interval at 95% limits of agreement between LiDAR-based and manual anthropometric measures of thigh length (**left**) and shank length (**right**). Length measures from the left and right leg of each participant (i.e., *n* = 16 thigh, *n* = 11 shank) were taken into account.

**Table 1 sports-12-00092-t001:** Average thigh and shank lengths of each participant from manual assessment compared to LiDAR-based assessment with *p*-values given from Student’s paired *t*-test, coefficient of determination (R^2^) from correlation analysis, as well as bias, upper and lower limit from the Bland–Altman analysis, respectively. For each participant (i.e., n = 16 thigh, n = 11 shank), the length measures of the left and right leg were combined for statistical analysis.

	Manual	LiDAR	*p*-Value	R^2^	Bias	Upper Limit	Lower Limit
Thigh length [cm] (*n* = 16)	45.8 (3.73)	45.0 (3.38)	0.203	0.901	0.21	2.92	−2.50
Shank length [cm] (*n* = 11)	40.3 (3.01)	38.2 (2.14) *	0.001	0.830	0.89	3.17	−1.40

* indicates a significant difference between ISAK and LIDAR-based assessment with *p* < 0.05.

## Data Availability

Data from this study are unavailable due to ethical restrictions.

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
