# Peer review of "Feasibility of Using Laser Imaging Detection and Ranging Technology for Contactless 3D Body Scanning and Anthropometric Assessment of Athletes"

_sports, 2024, doi:10.3390/sports12040092_

Round 1
Reviewer 1 Report
Comments and Suggestions for Authors
The authors aimed to assess the feasibility of using laser imaging detection and ranging technology for contactless 3D body scanning of athletes and deriving anthropometric measurements of lower limbs. Congratulations to the authors for this pilot study. Some issues and concerns should be addressed. For instance, the number of participants is too low to guarantee any measure of validity or repeatability. Specific comments are provided below.
- Please, provide a specific conclusion to answer the main purpose of the study in the abstract.
- Please, clarify purposes that are not consistent between the one presented in the abstract and the other presented in the introduction.
- Please, provide more data on methods and procedures to allow replication (i.e., order of analysis, randomization process, time periods, and specific variables that were used to compare anthropometrics. Also, the lengths were not analyzed in the 22 participants. So, groups should be specified.
- Manual vs. LiDAR revealed different values, although high agreement and R values. So, this can reveal reliability but not validity. Please, can you discuss this?
- Please, report appropriate conclusions according to the purpose.
Author Response
Response to reviewers’ comments
We would like to thank the reviewers for the valuable comments and overall positive feedback to our manuscript.
We addressed the specific comments of Reviewer 1 as follows:
- Abstract: We added the following specific conclusion to the abstract to highlight the main purpose of the study (Line 24-27): ‘The outcome of this pilot study is considered promising, and a further refinement of the proposed scanning protocol and advancement of iOS software for 3D reconstruction are recommended to exploit the full potential of the LiDAR technology for contactless anthropometric assessment of athletes.’
- Purpose: We clarified the purpose in the abstract and the introduction as recommended (Line 24-27, Line 55-63).
- Method: We added more information regarding the methodology and procedures to allow replication. Specifically, we have revised the manuscript and added the following information:
- Randomisation and order of analysis:
Data acquisition was not randomised. For all participants, the LiDAR scan was firstly obtained, followed by manual anthropometric measurements (Line 85-86). - Specific variables that were used to compare anthropometrics:
Measurements of thigh and shank length were extracted as study outcome parameters from the 3D models (Line 88-89). The comparison was limited to thigh and shank lengths measures due to the constraints of the Polycam software, which only allowed for linear measurements to be extracted (Line 95-97). - Also, the lengths were not analyzed in the 22 participants. So, groups should be specified:
Participants were categorized into three groups based on the quality of the LiDAR scan: 'poor' (n=6), 'moderate' (n=5), and 'good' (n=11). Definitions of these categories are detailed in the methods section (Line 90-93). Consequently, this categorization yielded 16 thigh length measurements (from the 'moderate' and 'good' groups, n=5+11) and 11 shank length measurements (from the 'good' group, n=11) (Line 116-118).
- Randomisation and order of analysis:
- Results and Discussion: We agree with Reviewer 1 that the present results only suggest a certain degree of reliability but not validity. We addressed this point in the discussion (Line 170-173) and conclusion (Line 189-191) accordingly. In particular, the goal of this pilot study was to assess the feasibility of using the LiDAR technology for contactless anthropometric assessment. The study design and sample size do not allow to make decisive conclusions regarding reliability or validity of the technology. To substantiate the present preliminary results with scientific rigor, further studies involving repeated measurements in a larger sample size under controlled conditions are recommended.
- Conclusion: We added a specific conclusion according to the purpose of the study to the final section (Line 185-189): ‘Despite challenges with inconsistent lighting across body parts and remaining limitations of available iOS software, the outcomes of this pilot study are considered promising. Further advancements of the proposed scanning protocol and available iOS software for 3D reconstruction are highly recommended to exploit the full potential of the LiDAR technology for contactless anthropometric assessment of athletes. For validation purposes, future studies should consider a repeated measurement design with larger sample sizes to substantiate the present preliminary results with scientific rigor.’
____________
Reviewer 2 Report
Comments and Suggestions for Authors
Thank you for the opportunity to revise the current MS. Here, there seem to be formatting issues or a wrong MS version was uploaded. Even so, please see my comments below, line by line:
Line 35: Formatting issue (reference)
Line 58: Could you please outline a study hypothesis here.
Line 61: Approval number is warranted here.
Line 85: Extra space, please amend.
Line 116: Table 1, what is this ?
Line 174: Any study limitations you might outline here ?
Comments on the Quality of English LanguageThank you for the opportunity to revise the current MS. Here, there seem to be formatting issues or a wrong MS version was uploaded. Even so, please see my comments below, line by line:
Line 35: Formatting issue (reference)
Line 58: Could you please outline a study hypothesis here.
Line 61: Approval number is warranted here.
Line 85: Extra space, please amend.
Line 116: Table 1, what is this ?
Line 174: Any study limitations you might outline here ?
Author Response
Response to reviewers’ comments
We would like to thank the reviewers for the valuable comments and overall positive feedback to our manuscript.
We addressed the specific comments of Reviewer 2 as follows:
- Hypothesis: We added a specific hypothesis to the Introduction as follows (Line 60-63): ‘It was hypothesized that the LiDAR technology allows for contactless measurement of shank and thigh length based on 3D body surface scanning in the training-specific setting with manual measurements according to ISAK standards as reference values.’
- Ethics approval: We added the specific approval number (Line 66).
- Limitations: We added the limitations of the study to the conclusion as follows (Line 185-189): ‘Despite challenges with inconsistent lighting across body parts and remaining limitations of available iOS software, the outcomes of this pilot study are considered promising. Further advancements of the proposed scanning protocol and available iOS software for 3D reconstruction are highly recommended to exploit the full potential of the LiDAR technology. For validation purposes, future studies should consider a repeated measurement design with larger sample sizes to substantiate the present preliminary results with scientific rigor.’
- Formatting & References: We have ensured consistent formatting throughout the manuscript and reduced the number of references to our own work as requested.
_________________________________________________
Again, we would like to thank the reviewers for their valuable feedback and comments. We hope that the revised manuscript complies with your expectations and is accepted for publication as short communication in Sports, Special Issue “Digital Technologies: Applications, Window of Opportunity and Challenges in Exercise, Health and Sports”.
Round 2
Reviewer 2 Report
Comments and Suggestions for Authors
I would like to thank the authors for following my suggestions. Still, MS formatting is still inappropriate for publication, so please handle this issue as soon as possible.
Comments on the Quality of English LanguageI would like to thank the authors for following my suggestions. Still, MS formatting is still inappropriate for publication, so please handle this issue as soon as possible.